# Characterization of Heterogeneous Microstructure and Mechanical Properties of Q345R Welded Joints including Roles of the Welding Process

**Qingkun He [1], Xiaodong Hu [2,*], Qitong Sun [2], Zhilong Dong [3], Xuefang Xie [3] and Han Wang [4]**

[1] College of Materials Science and Engineering, Shandong University of Science and Technology, Qingdao 266590, China
[2] College of Mechanical and Electronic Engineering, Shandong University of Science and Technology, Qingdao 266590, China
[3] College of New Energy, China University of Petroleum (East China), Qingdao 266580, China
[4] Anqiu Natural Resources and Planning Bureau of Shandong Province, Weifang 262100, China
* Correspondence: huxdd@163.com

**Abstract:** In this paper, the roles of the welding process on the heterogeneous microstructure and mechanical properties of Q345R welded joints were analyzed by a series of tests, including the preparation of welded joints with different welding processes, optical microscope observation, uniaxial tensile tests, and hardness and impact measurements. The experimental results show that with the increase in welding heat input, the content of pre-eutectoid ferrite and the size of the Weidner structure increased, while the hardness and the impact absorption energy of the weld zone decreased gradually. With the increase in heat input, the volume proportion of eutectoid ferrite in the weld increased from 9.90% to 18.78%; the volume proportion of acicular ferrite decreased from 85.10% to 76.21%. With the decrease in heat input, the volume proportion of eutectoid ferrite decreased from 10.58% to 1.45%, and the volume proportion of acicular ferrite increased from 84.21% to 92.74%. Under the influence of the second welding heat, the first weld zone, the fusion zone and part of the heat-affected zone were re-austenitizing, and the distribution of ferrite and pearlite was more uniform. The hardness value of the former weld was lower than that of the second weld, and the distribution was more uniform. The maximum hardness value of the second weld zone and its corresponding heat-affected zone increased with the increase in depth. The distribution of the yield strength and the tensile strength of welded joints was similar to that of hardness.

**Keywords:** welded joints; microstructure; mechanical properties; welding heat input

## 1. Introduction

Welding is the main technology in the manufacturing process of pressure vessels, and submerged arc welding is among the most widely used welding methods [1]. It is of great significance to study the microstructure of its welded joints through metallographic experiments, and it is particularly important to study the morphology and phase transformation process of ferrite. Ferrite can be divided into equiaxial ferrite, grain boundary ferrite, weistenite ferrite and massive ferrite according to morphology and transformation mechanism. Grain boundary ferrite is a relatively stable ferrite structure, which is commonly found in hypoeutectoid steels cooled slowly after welding. Weistenitic ferrite is a kind of unstable ferrite structure. Parallel weistenitic ferrite will maximize the deterioration of mechanical properties, and intersecting weistenitic ferrite can have good comprehensive mechanical properties [2]. Zhu, Guoxin et al. [3,4], for submerged arc welding in the practical application of research, show that submerged arc automatic welding can use the big electric current, wide range of welding line energy, high efficiency, relatively deep depth of molten weld quality, good working environment and high degree of automation in a

thick plate length in weld seam welding, etc. The weld zone and the heat-affected zone are too narrow to measure their strength through a tensile test. Hardness can be obtained from the test, and the distribution law of the yield strength and the tensile strength of welded joints can be obtained by using the relationship between hardness and strength [5,6]. Sun Dongsheng [7] studied the influence of electrode arc welding process parameters on the weld forming and microstructure properties of low-carbon steel and stainless steel. The research showed that the weld depth and the weld width of the two increase with the increase in electrode diameter and current. With the increase in the wire energy, the microstructure of the weld zone of low-carbon steel becomes coarse, and the microstructure and the hardness of the superheated zone become larger. The microstructure of a stainless steel welded joint increases with the increase in the wire energy but the hardness does not change. Xing Kui [8], through the straight seam submerged arc welding experiment of a steel plate, concluded that changing the welding line energy will change the type, shape, size and distribution of the microstructure component phase of the heat-affected zone (HAZ) of the welded joint. Min Xiaofeng [9] conducted an experiment on the influence of welding process parameters on the microstructure and properties of a welded joint of 9 Ni steel. The study showed that different welding process parameters change the microstructure of the joint. Under high-current and high-voltage conditions, the variable width of the heat-affected zone becomes larger, the joint plasticity becomes worse, and the hardness value of the heat-affected zone slightly decreases. Wang et al. [10] conducted Vickers hardness distribution test of welded joints, and the study showed that the hardness of the weld and the heat-affected zone of welded joints were higher than that of the base material. With the decrease in the welding line energy, the microhardness of the weld zone gradually increased, and the hardness of the heat-affected zone gradually decreased with the increase in distance from the weld.

Yang Li [11] conducted a study on the influence of the welding wire energy of submerged arc welding on the performance of a WEL TEN80A welded joint of high-strength steel. The study showed that with the increase in the wire energy, the microstructure of the overheated zone becomes coarse, the mechanical properties of the welded joint deteriorate, and the impact toughness decreases most obviously. Compared with a thick plate, a thin plate has poor thermal conductivity and is more sensitive to the line energy. Pan et al. [12] conducted DC and AC welding tests on 16 Mn steel with the same equipment, and the research showed that under AC and DC conditions, the two groups have the same microstructure composition and are composed of ferrite and pearlite. The elongation at break is lower than that of the base metal, and the impact toughness of the weld is better than that of the base metal. Because of the weistenite structure in the weld zone under AC conditions, the minimum impact toughness of the weld is lower than that in DC conditions.

It was found that low welding heat input can reduce welding deformation, obtain uniform, dense and refined microstructure, and improve welding joint performance [13]. Lala believed that when the welding methods were different, the level of heat input would have different influences on the hardness of the weld zone, and the hardness of the heat-affected zone would decrease with the increase in heat input [14]. Vedrtnam et al. [15] used RSW, regression equation and GA to study the influence of welding process parameters on the weld width, the weld height and the hardness of submerged arc automatic welding. The study showed that with the increase in welding voltage, the weld width increases and the weld height decreases; with the increase in current, the weld height increases and the weld width remains unchanged; the weld width and the weld height both decrease with the increase in welding speed; with the increase in NPD from nozzle to plate, the weld width decreases and the weld height increases; the weld hardness increases with the increase in current, and is not affected by voltage. Olena and Berdnikova et al. [16] studied the thermal input on the working performance of welded joints, and showed that the strength, ductility and crack resistance of welded joints of low-alloy high-strength steel is significantly improved with the reduction in the thermal input value.

## 2. Materials and Experiments

### 2.1. Preparation of Weld Joints

The evaluated plate was selected as the hot-rolled Q345R steel plate of 400 mm × 150 mm × 16 mm. All the test plates were cut from the same steel plate. Double-sided welding was carried out by the submerged arc automatic welding, and the welding direction was perpendicular to the rolling direction. The MZ1250 submerged arc automatic welding machine was used, and the welding material was H10Mn2 welding wire with φ4 mm and SJ101 flux. Table 1 shows the chemical composition of Q345R and H10Mn2 welding wires and welding wire–flux combination cladding metal. The temperature between layers was limited within the range of 100–150 °C. The groove size is shown in Figure 1, and the four sets of welding process parameters were considered, as shown in Table 2.

**Table 1.** Chemical compositions of Q345R steel and H10Mn2 and welding wire (wt. %).

| Material | C | Mn | Si | P | S | Ni | O | Ce | Fe |
|----------|-----|------|------|-------|--------|-------|-------|-------|-----|
| Q345R | 0.14 | 1.45 | 0.27 | 0.012 | 0.004 | 0.07 | 0.06 | 1.59 | Bal |
| H10Mn2 | 0.90 | 1.82 | 0.04 | 0.011 | 0.005 | <0.10 | <0.10 | 2.72 | Bal |
| H10Mn2+SJ101 | 0.069 | 1.63 | 0.29 | 0.019 | 0.0058 | 0 | 0.0044 | 1.699 | Bal |

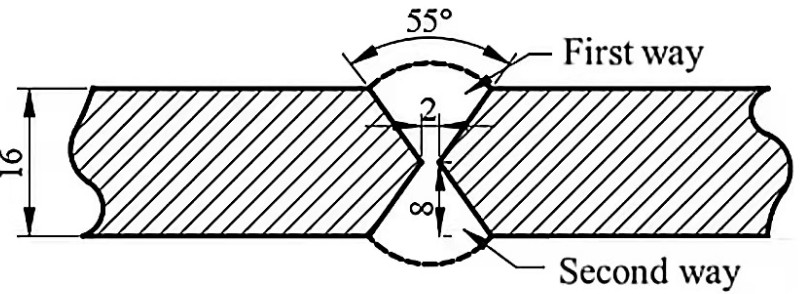

**Figure 1.** Schematic diagram of a Q345R welded joint (mm).

**Table 2.** Four sets of welding process parameters.

| Specimen ID | Welding Pass | Voltage/V | Current/A | Welding Speed (mm/s) | Heat Input (kJ/cm) | Wire Brand | Wire Diameter (mm) |
|-------------|--------------|-----------|-----------|----------------------|--------------------|------------|--------------------|
| 1# | First | 36 | 730 | 6.5 | 40.4 | | |
| | Second | 36 | 730 | 6.41 | 41.0 | | |
| 2# | First | 36 | 450 | 6.54 | 24.8 | | |
| | Second | 36 | 450 | 6.4 | 25.3 | H10Mn2 | 4 |
| 3# | First | 33 | 450 | 6.5 | 22.8 | | |
| | Second | 33 | 450 | 6.4 | 23.2 | | |
| 4# | First | 33 | 730 | 6.52 | 36.9 | | |
| | Second | 33 | 730 | 6.41 | 37.6 | | |

### 2.2. Microstructural Characterization

The distribution of the metallographic microstructure of weld joints was observed by the optical microscope according to the standard GB/T13298-2015 "Metal Microstructure Inspection Method". The small samples of 30 mm × 10 mm × 16 mm were cut from the weld joints by the TQY-06A slow-moving wire cutting machine. The corrosion solution was 4% nitric acid alcohol solution, and the corrosion time at room temperature was 5–10 s, until the surface brightness became dark. Additionally, the sampling locations are shown in Figure 2 to cover the weld zone, the fusion zone, the heat-affected zone and the base metal zone. As shown in Figure 3, the sampling locations of microstructures in the overheated zone and the weld zone of the four groups of test plates were the same locations 2 and 0.5 mm before and after the fusion line in the horizontal direction of h/2 of the second weld.

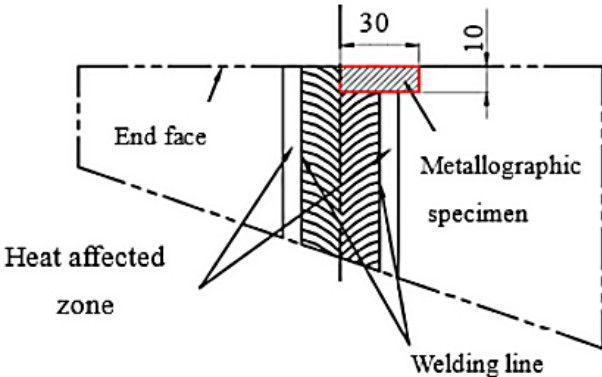

**Figure 2.** Sampling diagram of metallographic sample (mm).

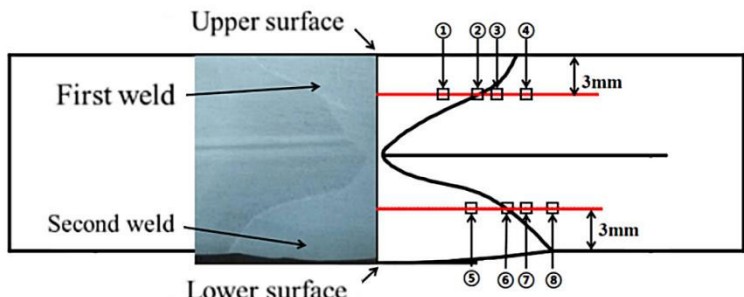

**Figure 3.** Sampling diagram of metallographic test for four groups of welded joints.

### 2.3. Mechanical Tests

The distribution of hardness across whole Q345R welded joints was characterized by the automatic micro-Vickers hardness tester FM-700+SVDM-4R with image processing at room temperature in accordance with the GB/T2654-2008 "Welding Joint Hardness Test Method". The working load and the holding time were 1.931 N and 15 s, respectively.

According to the standard GB/T2650-2008 "Welding Joint Impact Test Method", the impact sample was intercepted, as shown in Figure 4. According to GB/T229-2007 "Metal Materials Charpy pendulum impact test method", the IMPACT absorption energy of 12 specimens was measured by using the JBN-300 pendulum testing machine.

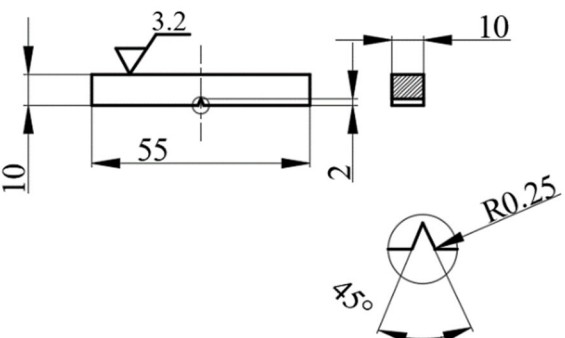

**Figure 4.** Schematic diagram of cutting position of impact sample (mm).

### 2.4. Uniaxial Tensile Tests

The distribution of mechanical properties was further characterized by the uniaxial tensile tests of the miniature specimen whose detailed dimensions are shown in Figure 5. Additionally, the sampling position of miniature tensile specimens are introduced in Figure 6, i.e., three groups of specimens were cut from the BM, HAZ and WM, respectively, and each group had two specimens. During the tensile tests, the displacement was controlled as 0.6 mm/min, while the imposed loads on the specimen were recorded by a force sensor. Finally, the fracture morphologies were also observed by the scanning electron microscope.

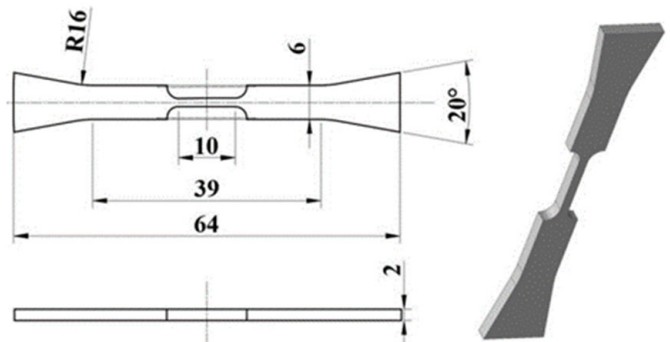

**Figure 5.** Detailed dimension of miniature tensile (mm).

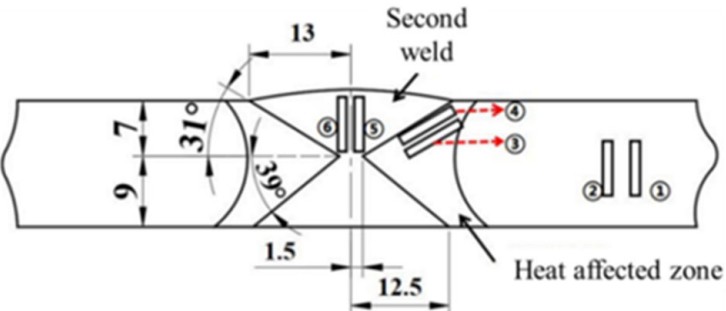

**Figure 6.** Schematic diagram of sampling position of miniature tensile specimen (mm).

*2.5. Welding Joint Toughness Tests*

In order to study the influence of different welding heat input on the impact toughness of the weld zone, three impact samples with the same specifications were prepared in four groups of the weld zone and impact tests were carried out. The results of the impact test are shown in Table 3. When the maximum welding heat input is 41.0 kJ/cm, the impact absorption energy is 92.7 J, and the impact toughness is the worst. When the welding heat input is 21.2 kJ/cm, the impact absorption energy is 140.7 J, and the impact toughness is the best. In the four groups of heat input, the higher the heat input, the worse the impact toughness of the weld zone. The reason is that with the increase in welding heat input, the proportion of proeutectoid ferrite in the weld zone increases, while the acicular ferrite content gradually decreases, making the impact toughness of the weld zone decrease.

**Table 3.** Impact toughness of the weld zone under different heat input.

| Test Plate Number | Sample | Shock Absorption Work (J) | Average Value (J) | Error (%) | Heat Input (kJ·cm$^{-1}$) |
|---|---|---|---|---|---|
| | The first sample | 92 | | 0.8 | |
| 1# test panel | The second sample | 94 | 92.7 | 1.4. | 41.0 |
| | The third sample | 92 | | 0.8 | |
| | The first sample | 136 | | 1 | |
| 2# test panel | The second sample | 130 | 134.7 | 3.5 | 25.3 |
| | The third sample | 138 | | 2.4 | |
| | The first sample | 140 | | 0.5 | |
| 3# test panel | The second sample | 142 | 140.7 | 0.9 | 23.2 |
| | The third sample | 140 | | 0.5 | |
| | The first sample | 95 | | 2.4 | |
| 4# test panel | The second sample | 98 | 97.3 | 0.7 | 37.6 |
| | The third sample | 99 | | 1.7 | |

## 3. Results and Discussion

### 3.1. Influence of Welding Heat Input on the Microstructure of a Welded Joint

The effect of the welding process on the microstructure formation of the weld zone is illustrated in Figure 7a–d. Under the four sets of welding parameters, the microstructures of the weld zone are all flaky and massive precipitating co-precipitated along the columnar grain boundary. Deformation of ferrite, and acicular ferrite that grows parallel to the crystal and a large amount of pearlite. It can be seen from the figure that with the increase in welding heat input, the grain size of proeutectoid ferrite increases, and the structure becomes obviously coarser and shorter, and the white areas of massive proeutectoid ferrite precipitated along the columnar crystal form gradually widening, the number of acicular ferrite gradually decreases, and the grain size increases to a certain extent.

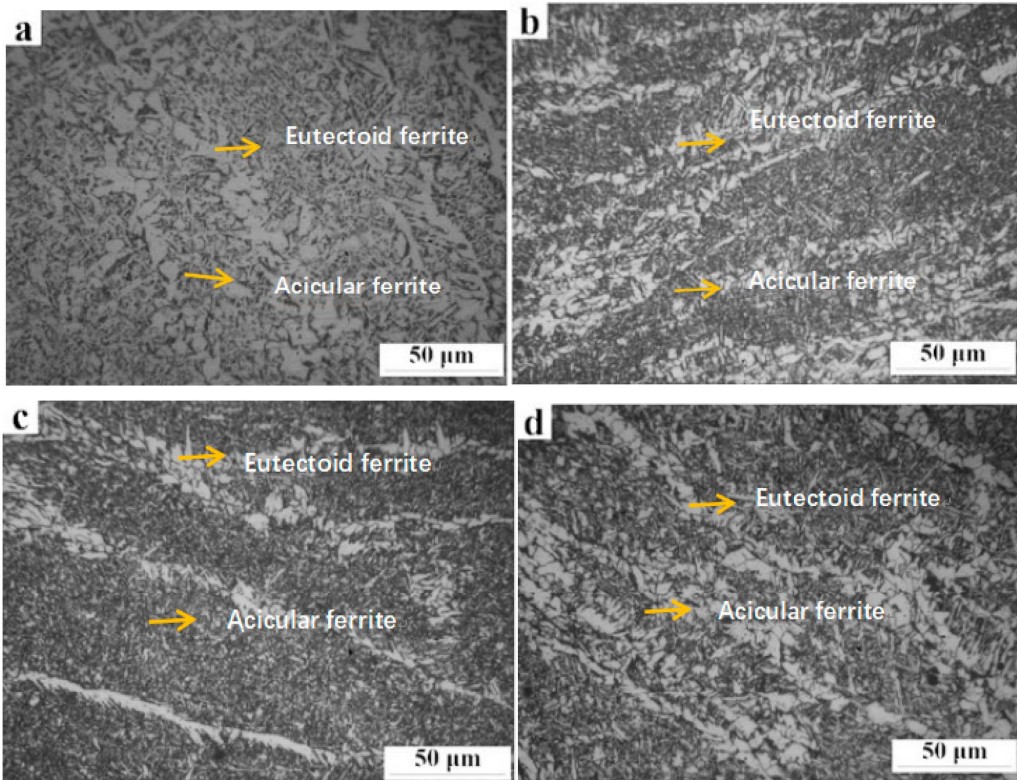

**Figure 7.** Microstructure of the weld zone under different heat inputs: (**a**) 41.0 KJ·cm$^{-1}$; (**b**) 25.3 KJ·cm$^{-1}$; (**c**) 23.2 KJ·cm$^{-1}$; (**d**) 37.6 KJ·cm$^{-1}$.

The microstructures of the superheated zone under different heat inputs are shown in Figure 8a–d. The microstructures of the four groups of the superheated zone are all Widmanstätten microstructures, namely proeutectoid ferrite and needles precipitated along the grain boundaries of prior austenite and in grained shaped ferrite and a small amount of pearlite. With the increase in welding heat input, the proeutectoid ferrite in the superheated zone appears to be distributed in a network along the original austenite grain boundary, and the Widmanstätten structure size increases significantly.

The weld zone: when the welding heat input is 23.2 KJ/cm (3#), the volume of proeutectoid ferrite accounts for 9.90%, and the volume of acicular ferrite accounts for 85.10%; When the heat input is 25.3 KJ/cm (2#) and 37.6 KJ/cm (4#), the volume proportion of proeutectoid ferrite is 13.51% and 15.96%, respectively, and the volume proportion of acicular ferrite is 81.25% and 79.47%, respectively; with the increase in heat input to 41.0 KJ/cm (1#), the volume proportion is as high as 18.78%, and the volume proportion of acicular ferrite decreases to 76.21%. The heat-affected zone: when the heat input is 41.0 KJ/cm (1#), the volume of proeutectoid ferrite in the overheated zone accounts for 10.58%, and the microstructure grain is the coarsest, and the volume of acicular ferrite

accounts for 84.21%. At 37.6 KJ/cm (4#) and 25.3 KJ/cm (2#), the volume proportion of proeutectoid ferrite is 8.17% and 1.45%, respectively, while the volume proportion of acicular ferrite is 87.06% and 92.74%, respectively.

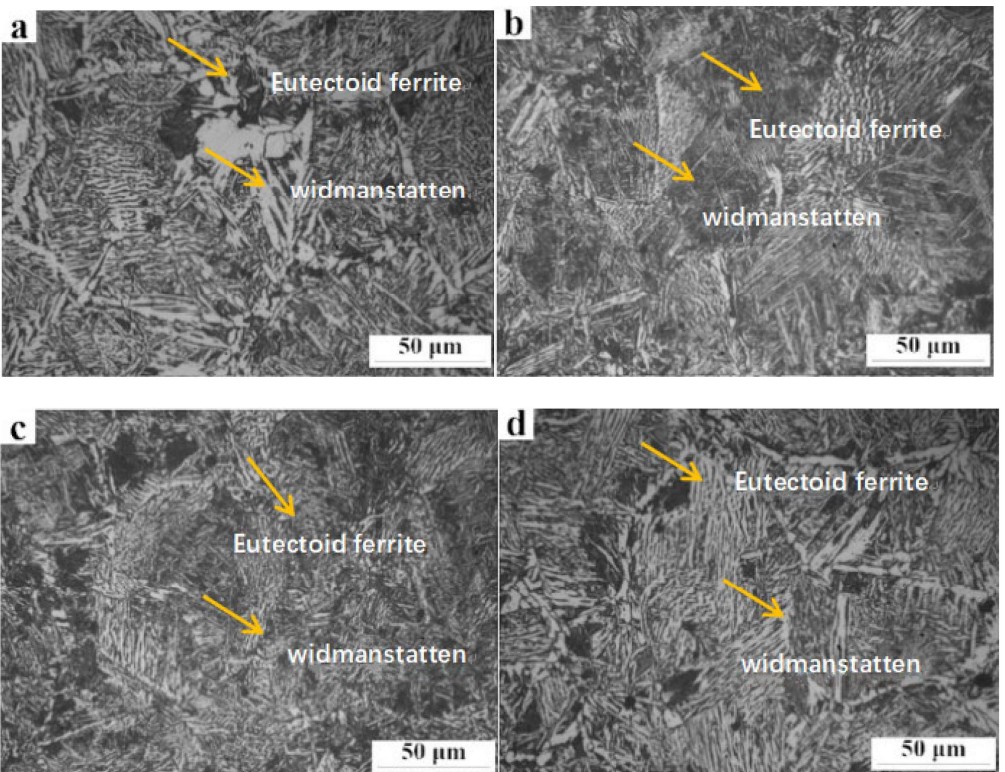

**Figure 8.** Microstructure of overheated zone under different heat inputs: (**a**) 41.0 KJ·cm$^{-1}$; (**b**) 25.3 KJ·cm$^{-1}$; (**c**) 23.2 KJ·cm$^{-1}$; (**d**) 37.6 KJ·cm$^{-1}$.

Figure 9 shows the position of the microstructure analysis of the welded joint. Figures 10 and 11 show the microstructure observed at the positions (1–4) and (5–8) of metallographic structure, namely, the weld zone 3 mm away from the two surfaces in the first and second channels, the fusion zone, the heat-affected zone 0.5 mm away from the fusion line, and the heat-affected zone 1.5 mm away from the fusion line.

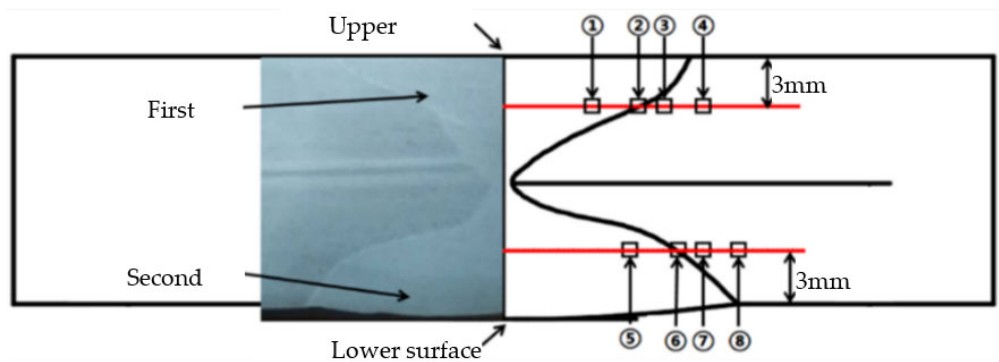

**Figure 9.** Distribution of sampling locations for metallographic microstructure observation.

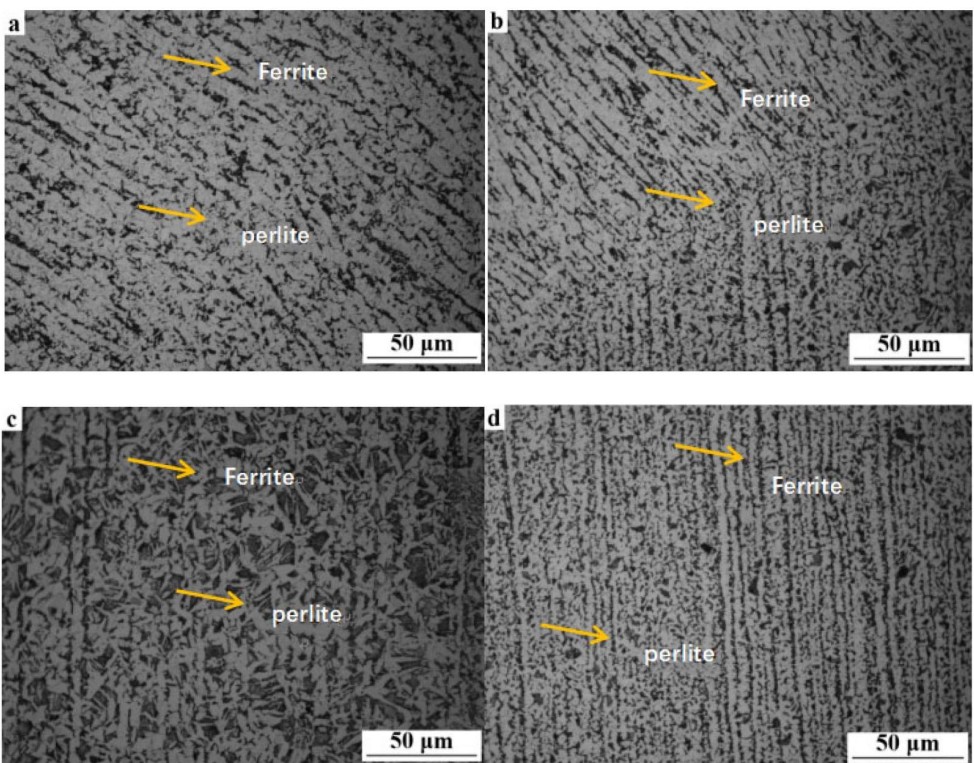

**Figure 10.** Microstructure of the first weld zone, the fusion zone and the heat-affected zone of 1–4, (**a**) the weld zone; (**b**) the fusion zone; (**c**) the heat-affected zone 0.5 mm from the fusion line: (**d**) the heat-affected zone 1.5 mm from the fusion line.

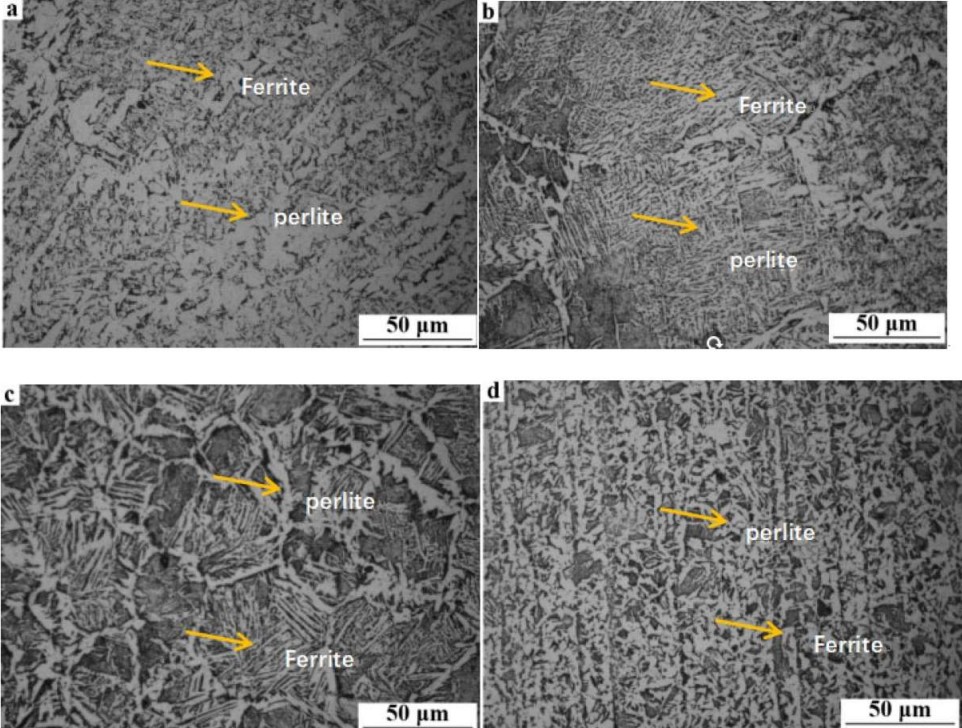

**Figure 11.** Microstructure of the first weld zone, the fusion zone and the heat-affected zone of 5–8, (**a**) the weld zone; (**b**) the fusion zone; (**c**) the heat-affected zone 0.5 mm from the fusion line; (**d**) the heat-affected zone 1.5 mm from the fusion line.

The distribution of microstructure for the 1# welded joints is firstly displayed in Figures 10 and 11. It can be seen that a large number of columnar crystal structures growing toward the center of the weld were observed at both two welding passes, which were formed during the rapid cooling, solidification and crystallization processes. The ferrite in the first seam increases, while the pearlite decreases. This is due to that under the influence of the second welding, the pearlite group decomposes into ferrite after a heat treatment in the first pass, and the ferrite grains grow up significantly and the distribution is more uniform. The Widmanstätten structure in the fusion zone and the heat-affected zone at a distance of 0.5 mm from the fusion line becomes pearlite and ferrite with a relatively uniform distribution under the influence of the second pass, and the Widmanstätten structure disappears. In the heat-affected zone at a distance of 1.5 mm from the fusion line, ferrite and pearlite are distributed in strips, and the structure is further refined, similar to the microstructure of the base metal. In addition, there are a lot of massive proeutectoid ferrite, acicular ferrite and part of pearlite around the columnar crystal in the second weld zone. The structure of the fusion zone and the heat-affected zone 0.5 mm away from the fusion line (that is, the overheated zone) is composed of massive and strip-like proeutectoid ferrite precipitated along the original austenite grain boundary, and needle-like intragranular growth. The Widmanstätten structure composed of ferrite and intragranular pearlite. This is because during welding, the temperature rises too fast, forming relatively coarse austenite grains, and then a special kind of overheating formed by a faster cooling rate organization. The microstructures in the heat-affected zone at a distance of 1.5 mm from the fusion line vary in size.

### 3.2. Hardness Distribution

In order to study the influence of welding parameters on the hardness of welded joints, the microhardness of welded joints of four groups of test plates was measured. As shown in Figure 12, hardness measurement points and lines are all horizontal lines perpendicular to h/2 of the second weld, and are measured from the fusion line to one side of the weld zone, the heat-affected zone and the base metal zone.

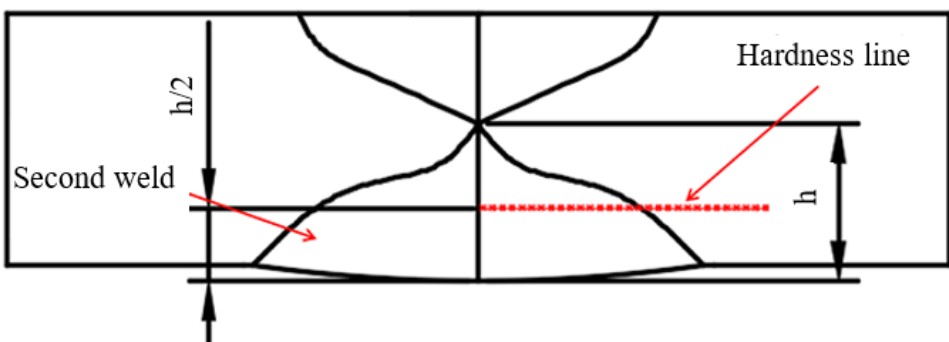

**Figure 12.** Schematic diagram of four groups of welding joint hardness measurement position.

The hardness distribution of welded joints under four groups of welding process parameters is shown in Figure 13. The hardness of the base metal is approximately 176 HV. The hardness of the weld zone decreases with the increase in the distance from the fusion line. When the distance from the fusion line exceeds 1–1.5 mm, the hardness of the weld zone reaches a stable state.

The hardness of the weld zone 1# and 4# is stable at approximately 188 HV, and the hardness of weld zone 2# and 3# is stable at approximately 215 HV. It can be seen that the hardness of the weld zone decreases with the increase in welding current, which is due to the gradual increase in the size of the proeutectoid ferrite in the weld zone. By comparing the hardness of weld zone 1# and 4#, 2# and 3#, it can be seen that when the current is constant and the voltage is changed in a small range, the hardness of the weld zone fluctuates little, and the hardness of the weld zone is mainly affected by the

current. The hardness of the heat-affected zone (HAZ) of the welded joints from 1# to 4# decreases gradually with the increase in the distance from the fusion line. The maximum hardness of the welded joints from 1# to 4# occurs in the overheated zone close to the fusion zone, because the brittle and hard Weidner structure is formed in the overheated zone. The maximum hardness of welded joint 1#, 2#, 3# and 4# is 226, 265, 272 and 231 HV, respectively. The maximum hardness of a welded joint decreases significantly with the increase in welding heat input and welding current. This is because with the increase in welding current and heat input, the grain size of Weistenite increases gradually, the acicular ferrite decreases, and the hardness value of the heat-affected zone decreases seriously. The increase in welding voltage has little effect on the hardness of the weld and the heat-affected zone, and the peak hardness decreases by approximately 5 HV.

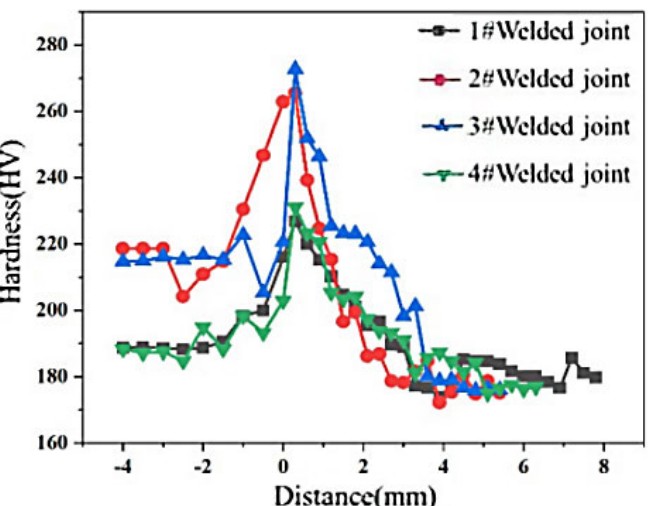

**Figure 13.** Hardness distribution of welded joints.

The distribution of measuring points was along 6 lines (L1–L6) which were horizontal and from the fusion line to the side of the weld zone, the heat-affected zone and the base material zone, as shown in Figure 14, and 60 equidistant points were measured for each line.

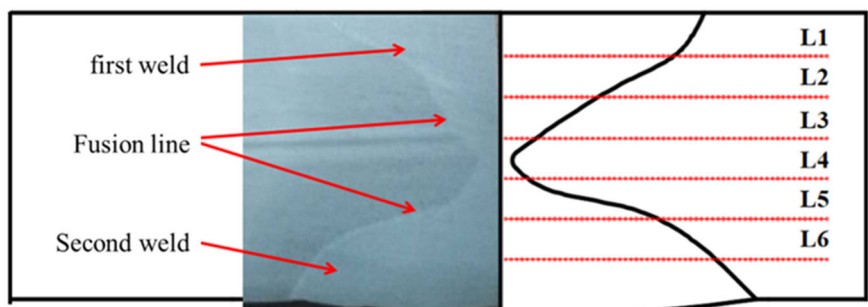

**Figure 14.** Distribution of measuring points for the hardness.

Figure 15 gives the distribution of hardness across the 1# welded joint. Obviously, the hardness of the first weld and the heat-affected zone is relatively uniform. The hardness of the weld zone and the heat-affected zone is stable at 184–198 HV, and the average hardness is 189.2 HV. The hardness of the second weld zone increases slightly with the increase in depth. The average hardness of the weld zone on the L4, L5, and L6 lines are 199.3, 198.2, and 195.6 HV, and the maximum hardness of the heat-affected zone is 218, 214, and 207 HV. Under the action of the welding thermal cycle, the next weld is equivalent to a heat treatment of the previous weld, so that the acicular ferrite crystals of the first weld and the Widmanstätten structure in the superheated zone disappear, and the residual stress of the

welded joint is reduced. So that the average hardness of the first weld is 10–20 HV lower than the hardness of the second weld.

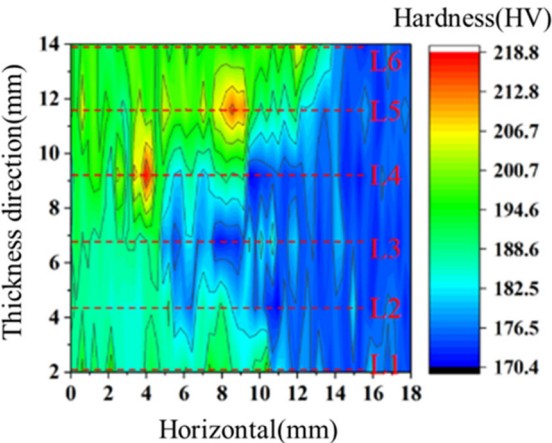

**Figure 15.** Distribution of hardness across the whole 1# welded joint.

### 3.3. Distribution Rule of Welding Joint Toughness

After the impact test, the fracture morphology of the four groups is shown in Figure 16. The fracture morphology of weld zone 1#, 2#, 3# and 4# shows the dissociation surface with river pattern distribution, namely cleavage fracture. However, there are a few dimples on the surface of weld zone 2# and 3#, with toughness stronger than that of weld zone 1# and 4#.

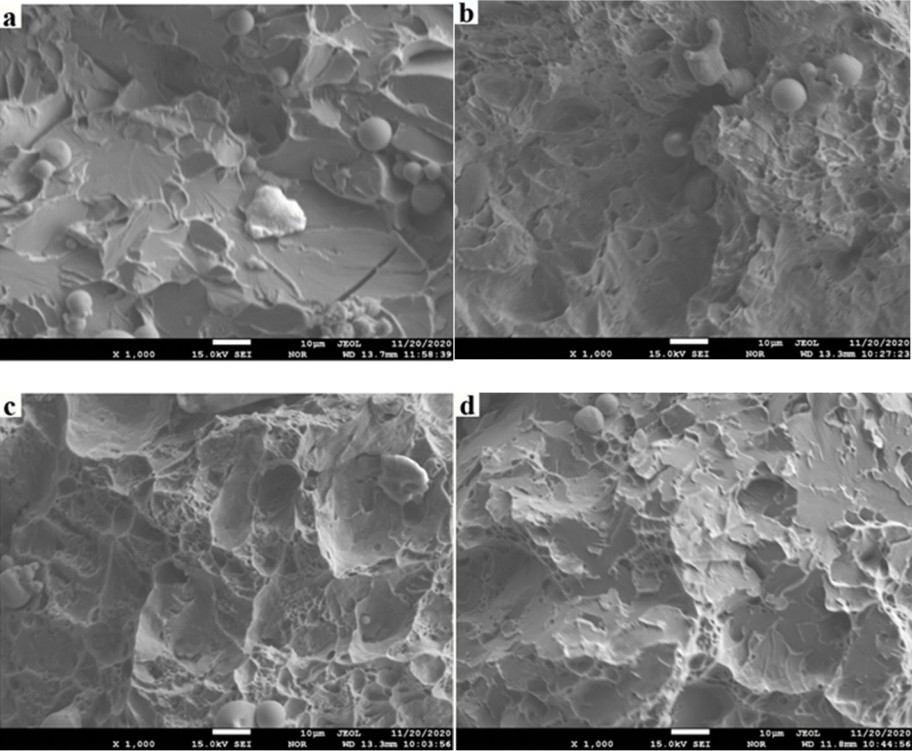

**Figure 16.** Impact fracture morphology of four groups of the weld zone, (**a**) 1# weld zone; (**b**) 2# weld zone; (**c**) 3# weld zone; (**d**) 4# weld zone.

The impact samples of the four groups of the weld zone are taken from the second seam. Figure 17 shows the relationship between the impact absorption work of the weld zone and the average hardness value of the weld zone under the four groups of heat

input: the impact toughness and the hardness of the weld zone gradually decrease with the increase in welding heat input.

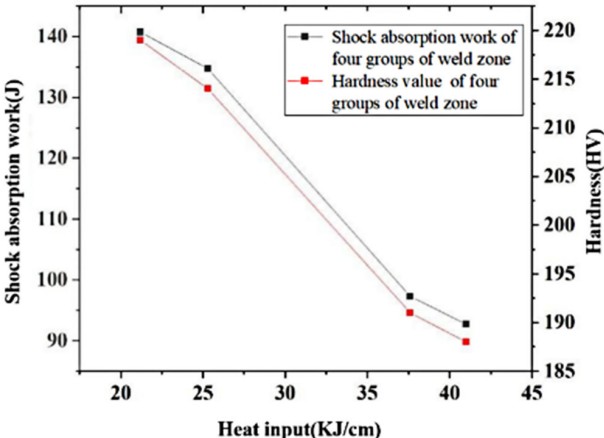

**Figure 17.** Variation of impact absorbed energy and hardness of the weld zone with heat input.

Notch positions of 10 groups of impact samples were cut at 2.5 mm from the lower surface of the 1# test plate, as shown in Figure 18. Notch positions of ①–③ were located at the weld zone 8, 5 and 3 mm away from the fusion line, notch positions of ④ were located at the fusion line, notch positions of ⑤–⑦ were located at 1, 2 and 3 mm away from the fusion line in the heat-affected zone. The ⑧–⑩ notch is located at the base metal. Furthermore, the distribution of impact absorption energy and the hardness along the L6 are shown in Figure 19. It can be seen that the greater the hardness value of the welded test board, the higher the hardness, the lower the impact absorption energy, the worse the impact toughness.

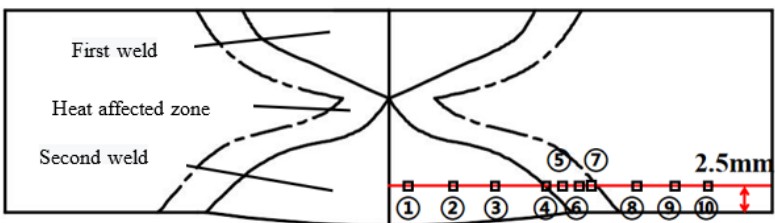

**Figure 18.** Schematic diagram of impact specimen.

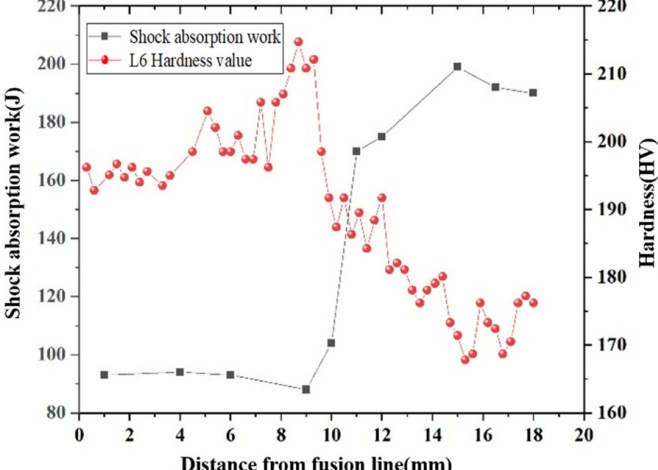

**Figure 19.** Distribution of impact absorbed energy and hardness along L6 for 1# weld joints.

Table 4 shows the impact absorption energy results of 10 groups of impact samples on the 1# welded test plate. When the notch of the impact specimen is opened at the fusion line, the impact absorption energy of the welded joint is the lowest 92 J, and the impact toughness is the worst. In the heat-affected zone, as the distance from the fusion line increases, the impact absorption energy gradually decreases, that is, the toughness decreases. When the notch of the impact specimen is opened in the weld area whose distance from the fusion line is greater than 3 mm, the impact absorption energy does not change much, and the impact absorption energy of the weld area is stable at approximately 108 J, that is, the impact toughness of the weld area farther from the fusion line is relatively stable. When the notch of the impact sample is opened on the base material, the impact absorption energy is the largest, that is, the position of the best impact performance of the test plate is in the base material area, and the impact absorption energy of the base material area is approximately 200 J.

**Table 4.** Impact absorption energy of 10 groups of 1# test plate.

| Gap Position | 1 | 2 | 3 | 4 | 5 | 6 | 7 | 8 | 9 | 10 |
|---|---|---|---|---|---|---|---|---|---|---|
| Impact absorbing energy/J | 108 | 108 | 106 | 92 | 114 | 180 | 185 | 200 | 202 | 199 |

*3.4. Tensile Properties*

The stress–strain curves obtained from the six sets of micro-tensile tests are shown in Figure 20 and the values are shown in Table 5. The six sets of micro-tensile curves have the same changing trend, and the yield strength and tensile strength obtained from the six sets of micro-tensile tests. The maximum values of 403.6 and 586.5 MPa appear in the stress–strain curve of the micro-tensile specimen labeled HAZ2, and the yield strength and the tensile strength of the stress–strain curve obtained from the micro-tensile specimen of the base material. The strength is less than that of the micro-tensile specimens taken from the weld and the heat-affected zone. As the distance from the fusion line increases, the yield strength and tensile strength obtained from the HAZ1 stress–strain curve are less than the yield strength obtained from the HAZ2 stress–strain curve. Therefore, the maximum value of the yield strength and the tensile strength of the welded joint should be near the fusion zone.

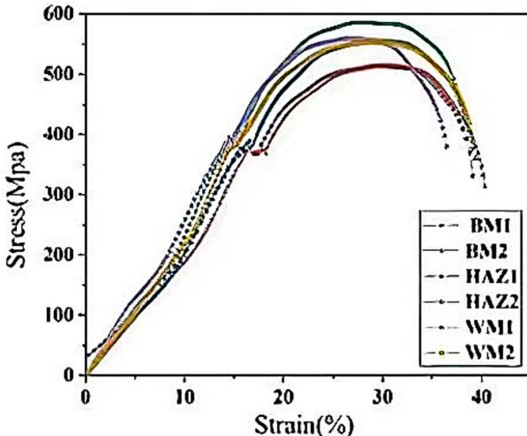

**Figure 20.** Micro-tensile stress–strain curve.

**Table 5.** Yield strength and tensile strength of micro tensile specimen.

| | BM1 | BM2 | HAZ1 | HAZ2 | WM1 | WM2 |
|---|---|---|---|---|---|---|
| Yield Strength/MPa | 376.2 | 374.9 | 383.8 | 403.6 | 387.6 | 383.6 |
| Tensile Strength/MPa | 513.9 | 515.1 | 554.6 | 586.5 | 559 | 553.1 |

In order to further study the size and distribution of the yield strength and the tensile strength of welded joints, and because the hardness of Q345R has a corresponding relationship with the yield strength and tensile strength, the yield strength and the tensile strength of Q345R can be calculated according to the micro-hardness measured in the previous section.

According to GB/T28896-2012 [17], the relationship between yield strength $\sigma_y$ and microhardness in the Q345R base metal region can be obtained as shown in Equation (1). Similarly, the tensile strength $\sigma_b$ can be obtained by microhardness conversion according to Equation (2) proposed by Kloos-Vel-ten [18]. Pargeter [19] proposed that the relationship between the yield strength $\sigma_y$ of the weld zone and microhardness can be expressed by Equation (3). Similarly, La van [20] proposed Equation (4) based on the relationship between the tensile strength $\sigma_b$ of the weld zone and microhardness. Yang verified the accuracy of Equations (1)–(4) by conducting SAW double-sided welding on 20 mm thick Q345R and conducting a micro-tensile test and a micro-hardness test. The relationship between yield strength $\sigma_y$, tensile strength $\sigma_b$ and microhardness in the heat-affected zone can be expressed by Equation (1) and Equation (4), respectively [21].

$$\sigma_y = 3.28HV - 211 \tag{1}$$

$$\sigma_b = 3.29HV - 47 \tag{2}$$

$$\sigma_y = 3.15HV - 168 \tag{3}$$

$$\sigma_b = 2.84HV + 28 \tag{4}$$

where $\sigma_y$ is the yield strength; $\sigma_b$ is tensile strength; $HV$ is Vickers hardness.

The yield strength and tensile strength distribution of the 1# welded joint are calculated from Equation (1) to Equation (4), as shown in Figures 21 and 22, respectively. The maximum yield strength and the tensile strength of the Q345R welded joint are 497 and 673 MPa at the fusion line, respectively. The minimum yield strength and tensile strength of the 1# welded joint are 353.5 and 516 MPa, respectively, at the base metal. The errors of yield strength and tensile strength (360.9 and 514.9 MPa) measured by a standard round rod tensile test of the same base material are 2.05% and 0.21%. The errors of yield strength and tensile strength (363.8 and 526.7 MPa) measured in the standard round rod tensile test of the same # welded joint are 2.91% and 2.07%, and the errors of yield strength and tensile strength (375.5 and 514.5 MPa) measured in the micro-tensile test of the same base material are 5.85% and 0.29%. The reliability of the micro-tensile test and the accuracy of Equations (1)–(4) are verified, indicating that the above equations can be used for automatic submerged arc welding.

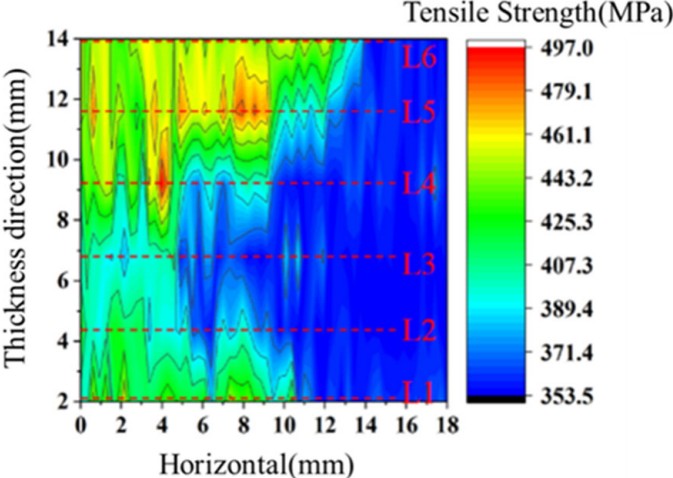

**Figure 21.** Contour of the yield strength of the 1# welded joint.

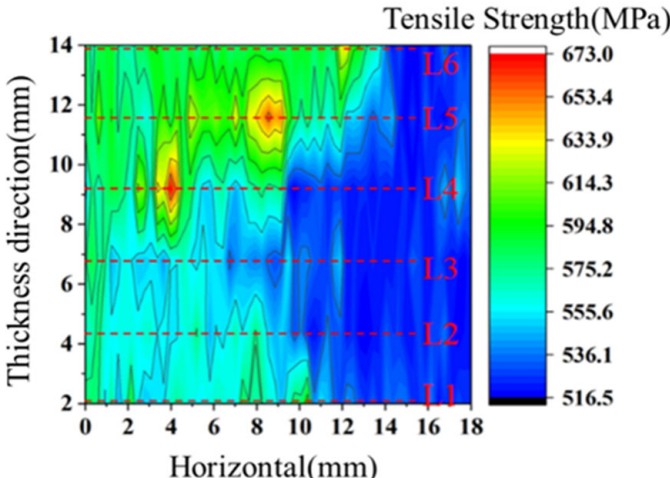

**Figure 22.** Contour of the tensile strength of the 1# welded joint.

The yield strength [22] and the tensile strength of the first weld and the heat-affected zone are evenly distributed. The maximum yield strength and tensile strength of the second weld appeared near the fusion zone, and the maximum yield strength [23] and tensile strength increased with the increase in depth. In addition, the yield strength and the tensile strength of the weld zone and the heat-affected zone [24,25] are evenly distributed except for the fusion zone and the overheated zone.

## 4. Conclusions

(1)    The microstructure of the weld zone consists of flake and massive pre-eutectoid ferrite precipitated along the columnar grain boundary and acicular ferrite and a large amount of pearlite growing in parallel to the grain. With the increase in welding heat input, the grain size of the pre-eutectoid ferrite in the weld zone increases. The white area of the massive pre-eutectoid ferrite becomes wider gradually, the number of acicular ferrite decreases and the grain size increases to a certain extent. With the increase in heat input, the volume proportion of eutectoid ferrite in the weld increased from 9.90% to 18.78%; the volume proportion of acicular ferrite decreased from 85.10% to 76.21%. With the decrease in heat input, the volume proportion of eutectoid ferrite decreased from 10.58% to 1.45%, and the volume proportion of acicular ferrite increased from 84.21% to 92.74%.

(2)    The hardness of the weld zone decreases with the increase in welding current. Because with the increase in welding heat input, the proportion of proeutectoid ferrite in the weld zone increases, while the acicular ferrite content gradually decreases, making the impact toughness of the weld zone decrease. The hardness of the heat-affected zone of the welded joint decreases gradually with the increase in the distance from the fusion line. The maximum hardness of the welded joint occurs in the overheated zone near the fusion zone. The maximum hardness of the welded joint decreases significantly with the increase in the welding heat input and welding current.

(3)    The impact toughness and the hardness of the weld zone decrease gradually with the increase in welding heat input. In the fusion zone, the impact absorption energy is at least 92 J, the toughness is the worst, and the impact is in the base material zone. At approximately 200 J of absorbed work, the toughness is the best.

(4)    All the three tensile tests fracture at the base metal, and the yield strength and the tensile strength of the welded joint are higher than that of the base metal, which means super match. To the side of the base material zone, with the increase in the distance from the fusion zone, the lower the yield strength and tensile strength until the base material zone reaches a stable state.

**Author Contributions:** Conceptualization, Q.H. and X.H.; methodology, X.H.; software, Q.S.; validation, Z.D., Q.S. and Q.H.; formal analysis, H.W; investigation, X.X.; resources, X.H.; data curation, H.W.; writing—original draft preparation, Q.H.; writing—review and editing, Q.S.; visualization, H.W.; supervision, Z.D.; project administration, Z.D.; funding acquisition, X.X. All authors have read and agreed to the published version of the manuscript.

**Funding:** This research received no external funding.

**Data Availability Statement:** Not applicable.

**Conflicts of Interest:** The authors declare no conflict of interest.

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
