# Peer review of "Characterization of Heterogeneous Microstructure and Mechanical Properties of Q345R Welded Joints including Roles of the Welding Process"

_metals, doi:10.3390/met12101708_

Round 1

Author Response

Detailed Response to Reviewers' Comments

Title: Characterization of Heterogeneous Microstructure and Mechanical Properties of Q345R Welded Joints Including Roles of Welding Process

Reviewer Comments

Reviewer #1

This work discussed the influence of the welding process on the heterogeneous microstructure and mechanical properties of Q345R welded joints. It was well written and the discussions were delivered properly. However, several minor clarifications and improvements can be made:

  1. The reviewer can understand the aim/goal of this work, however, it was not mentioned clearly in the introduction part. The reviewer suggested, it should be emphasized and written properly in the last part of the introduction.

R: Thanks to the reviewer's suggestion.

This paper studies the influence of heat input on the welding structure, hardness, toughness and mechanical properties through experiments. Relevant conclusions are drawn.

The microstructure of the weld zone consists of flake and massive pre-eutectoid ferrite precipitated along the columnar grain boundary and acicular ferrite and a large amount of pearlite growing in parallel to the grain. With the increase of welding heat input, the grain size of the pre-eutectoid ferrite in the weld zone increases. The white area of the massive pre-eutectoid ferrite becomes wider gradually, the number of acicular ferrite decreases and the grain size increases to a certain extent.

The hardness of the weld zone decreases with the increase of welding current. Because with the increase of welding heat input, the proportion of proeutectoid ferrite in the weld zone increases, while the acicular ferrite content gradually decreases, making the impact toughness of the weld zone decrease. The hardness of the heat affected zone of the welded joint decreases gradually with the increase of the distance from the fusion line. The maximum hardness of the welded joint occurs in the overheat zone near the fusion zone. The maximum hardness of the welded joint decreases significantly with the increase of the welding heat input and welding current.

The impact toughness and hardness of the weld zone decrease gradually with the increase of welding heat input. In the fusion zone, the impact absorption energy is at least 92J, the toughness is the worst, and the impact is in the base material zone. About 200J of absorbed work, the toughness is the best.

All the three tensile tests fracture at the base metal, and the yield strength and tensile strength of the welded joint are higher than that of the base metal, which means super match. To the side of the base material zone, with the increase of the distance from the fusion zone, the lower the yield strength and tensile strength until the base material zone reaches a stable state.

  1. In Figures 7 and 8, the reviewer can barely see the difference between two microstructures, more clarity images are needed perhaps.

R: Thanks to the reviewer's suggestion.The white part is the Eutectoid ferrite, and the black part is the Acoustic ferrite.

  1. The units in the caption for Figures 7 and 8 are also not consistent with the manuscript, please make it unified. The same problem goes for units presented on page 6 (second paragraph).

R:Thanks to the reviewer's suggestion.The problems of relevant units have been corrected in the original text.

  1. In Figure 10, please check the spelling for perlite.

R:Thanks to the reviewer's suggestion. The relevant modifications are as follows and have been corrected in the original text

Figure 10. Microstructure of the first weld zone, fusion zone and heat affected zone of 1-4, (a) Weld zoneï¼›(b) Fusion zoneï¼›(c) Heat affected zone 0.5mm from fusion lineï¼›(d) Heat affected zone 1.5mm from fusion line.

Figure 11. Microstructure of the first weld zone, fusion zone and heat affected zone of 5-8, (a) Weld zoneï¼›(b) Fusion zoneï¼›(c) Heat affected zone 0.5mm from fusion lineï¼›(d) Heat affected zone 1.5mm from fusion line.

  1. It will be great if the authors can unify all images’ font style and size to be nearly the same as in the paragraphs.

R:Thanks to the reviewer's suggestion.Relevant issues have been improved.

  1. The total number of references of 21 is quite small for this work. Adding more related works will increase the credibility of this study.

R:Thanks to the reviewer's suggestion. More references have been added.

Reviewer 2 Report

This paper presented a laboratory experimental results of welded joint of Q345R materials. The manuscript has merits, however, it requires revision before acceptance.

Abstract: is well written.

Introduction: It needs a massive reform. It does not justify the need for the research. It appears to a collection of summaries of open literature. There is no logical sequences of arguments that leads to the need for this research. No clear indication of the research gap on the topic is given. Please modify this.

Experimental: This section is the critical one. They were presented in a very confusing way. For example, if a reader looked at the Table 1, it will appear that three different material were welded. However, the truth is only one material was joined. The other material is welding rod and the final material is welded rod with flux. It made it very difficult for the reader to follow the experimental procedure.

Additionally, Table 2: Welding rod diameter is always 4mm. Why they need to mentioned in each of the lines? Please merge them. Also the wire band. Merge them. Additionally, when the flux was used. Why did you mention three materials in Table1 and why only two appears in Table 2. Please restructure completely the experimental section by clearly mentioning what was the different conditions were experimented. It was also not clear the reason behind the chosen heat input. Why the highest was 41 and why the lowest was 22? Why they do not appear in the chronological order? Why the voltage was different in different cases? Which standard was chosen for the mechanical test?

Result and discussion: All the picture should be improved. The quality of each of them is very poor. Why red mark appeared in the text inside each of the figures? A lot of spelling mistake is appeared in the text of each figures. There was no clear microstructural changes appeared in figures 7-11 depending on the heat input. All the changes are for different zones. The discussion section seems unnecessary long and repetition of similar figures. Just need to show one indicative figure and tell the reader that similar microstructural properties were observed for all conditions.

Why Table 4 appear suddenly? It should be presented in the experimental section that these are the conditions that were tested.

Author Response

Detailed Response to Reviewers' Comments

Title: Characterization of Heterogeneous Microstructure and Mechanical Properties of Q345R Welded Joints Including Roles of Welding Process

Reviewer Comments

Reviewer #2 

This paper presented a laboratory experimental results of welded joint of Q345R materials. The manuscript has merits, however, it requires revision before acceptance.

Abstract: is well written.

Introduction: It needs a massive reform. It does not justify the need for the research. It appears to a collection of summaries of open literature. There is no logical sequences of arguments that leads to the need for this research. No clear indication of the research gap on the topic is given. Please modify this.

R: Thanks to the reviewer's suggestion.The importance of this study is reflected in the influence of different heat inputs to study the microstructure and mechanical properties of the weld zone, heat affected zone and base metal zone after welding, so as to find the best heat input and improve the quality after welding.

Experimental: This section is the critical one. They were presented in a very confusing way. For example, if a reader looked at the Table 1, it will appear that three different material were welded. However, the truth is only one material was joined. The other material is welding rod and the final material is welded rod with flux. It made it very difficult for the reader to follow the experimental procedure.

R: Thanks to the reviewer's suggestion.This is really something that needs to be improved. Welding is classified as a kind of material and needs to be improved later.

Additionally, Table 2: Welding rod diameter is always 4mm. Why they need to mentioned in each of the lines? Please merge them. Also the wire band. Merge them. Additionally, when the flux was used. Why did you mention three materials in Table1 and why only two appears in Table 2. Please restructure completely the experimental section by clearly mentioning what was the different conditions were experimented. It was also not clear the reason behind the chosen heat input. Why the highest was 41 and why the lowest was 22? Why they do not appear in the chronological order? Why the voltage was different in different cases? Which standard was chosen for the mechanical test?

R:Thanks to the reviewer's suggestion. The electrode diameter of 4mm and other contents to be merged have been merged into one cell. One of the three materials is welding rod, so it is not shown in Table 2. The experiments under different conditions are different welding heat input, the maximum value is 41, and the minimum value is 22, which are the corresponding use values given by the company. Different voltages are used to change the heat input. The standard used for mechanical testing is the pressure vessel industry standard GB150-2011.

Result and discussion: All the picture should be improved. The quality of each of them is very poor. Why red mark appeared in the text inside each of the figures? A lot of spelling mistake is appeared in the text of each figures. There was no clear microstructural changes appeared in figures 7-11 depending on the heat input. All the changes are for different zones. The discussion section seems unnecessary long and repetition of similar figures. Just need to show one indicative figure and tell the reader that similar microstructural properties were observed for all conditions.

R:Thanks to the reviewer's suggestion. The errors in the relevant pictures have been corrected in the original text. There is no obvious microstructure change in Figure 7-11, which is corrected with an indicative number.

Figure 10. Microstructure of the first weld zone, fusion zone and heat affected zone of 1-4, (a) Weld zoneï¼›(b) Fusion zoneï¼›(c) Heat affected zone 0.5mm from fusion lineï¼›(d) Heat affected zone 1.5mm from fusion line.

Figure 11. Microstructure of the first weld zone, fusion zone and heat affected zone of 5-8, (a) Weld zoneï¼›(b) Fusion zoneï¼›(c) Heat affected zone 0.5mm from fusion lineï¼›(d) Heat affected zone 1.5mm from fusion line.

Why Table 4 appear suddenly? It should be presented in the experimental section that these are the conditions that were tested.

Rï¼›Thanks to the reviewer's suggestion. Table 4 should indeed be explained in the experimental part, and relevant problems have been corrected in the paper.

Reviewer 3 Report

Dear Authors,  

I have reviewed your paper titled: "Characterization of Heterogeneous Microstructure and Mechanical Properties of Q345R Welded Joints Including Roles of Welding Process".

The paper fulfils the aims and scope of Metals journal, and can be considered for potential publication. However, it needs some improvements. I have some suggestions, which are listed below. 

General remarks:

- Please add the quantitative results into the abstract.

- You have presented 22 references. Only one has been published in last three years. The science made big step forward last years. Please support your work with newly published  references.

Introduction:

- This section should be supported with newly published references. Moreover, I cannot find any information about weldability of used materials. You should support this section with some information about welding Q345R steel.

- You should state, what is the aim of your work. Moroever, the novelty has not been underlined here.

Experimental:
- I propose to cheng the name to "Materials and Experimental", because you have described materials too.

- Table 1 - the source of presented values is unknown. Have you analyzed these compositions? If yes, please mark used methods. If values were taken from other source (standard, paper, manufacturer datas), please add this information into the paper.

- From the weldability point of the view, the most important is carbon equivalent (Ce) of used steel. Please add this information into the table 1.

- Please show general mechanical properties of used materials - yield point, tensile strength and elongation.

- Figure 1 - "first way", "second way" - please use welding engineering terminology.

- Table 2 - following most standards, the heat input should be presented as "kJ/mm". Moreover, following this table, all beads were performed by the same wire, so, why in table 1, two wires were presented. Please improve. The information about diameter could be presented in the text.

Results and Dicussion:

- I cannot find any scientific discussion here. You have not compared your results with other scientific papers. It allows to underline the biggest advantages of your work, which could be used to mark the novelty and necessity of your work. Please add this kind of comparison.

- "Widmanstatten" - please change to proper name: "Widmanstätten"

- Have you observed any welding imperfections? If not, please underline this issue in the text. It proved that proposed parameters were good from technological point of the view.

- What kind of hardness scale was used, HV5, HV10...? Please mark in the paper.

- Fig. 13 - please show the most important values.

- Table 4- the name of table is "Impact toughness". However, I cannot find impact strength in the table.

Conclusions:

- Please support conclusions with the quantitative results.

Author Response

Detailed Response to Reviewers' Comments

Title: Characterization of Heterogeneous Microstructure and Mechanical Properties of Q345R Welded Joints Including Roles of Welding Process

Reviewer Comments

Reviewer #3 

Dear Authors,

I have reviewed your paper titled: "Characterization of Heterogeneous Microstructure and Mechanical Properties of Q345R Welded Joints Including Roles of Welding Process".

The paper fulfils the aims and scope of Metals journal, and can be considered for potential publication. However, it needs some improvements. I have some suggestions, which are listed below.

General remarks:

- Please add the quantitative results into the abstract.

- You have presented 22 references. Only one has been published in last three years. The science made big step forward last years. Please support your work with newly published references.

R: Thanks to the reviewer's suggestion.Relevant quantitative results have been added to the abstract to, and relevant recent references have been added to the article.

Introduction:

- This section should be supported with newly published references. Moreover, I cannot find any information about weldability of used materials. You should support this section with some information about welding Q345R steel.

- You should state, what is the aim of your work. Moroever, the novelty has not been underlined here.

R: Thanks to the reviewer's suggestion.The purpose of relevant work has been added to the introduction, and the novelty is also reflected.

Experimental:
- I propose to cheng the name to "Materials and Experimental", because you have described materials too.

- Table 1 - the source of presented values is unknown. Have you analyzed these compositions? If yes, please mark used methods. If values were taken from other source (standard, paper, manufacturer datas), please add this information into the paper.

- From the weldability point of the view, the most important is carbon equivalent (Ce) of used steel. Please add this information into the table 1.

- Please show general mechanical properties of used materials - yield point, tensile strength and elongation.

- Figure 1 - "first way", "second way" - please use welding engineering terminology.

- Table 2 - following most standards, the heat input should be presented as "kJ/mm". Moreover, following this table, all beads were performed by the same wire, so, why in table 1, two wires were presented. Please improve. The information about diameter could be presented in the text.

R: Thanks to the reviewer's suggestion.Thanks for the suggestion of experts, the name has been changed to "Materials and Experiments" in the paper. The data in Table 1 comes from the test data of enterprises, specifically Shandong Baite Group. The carbon equivalent (Ce) of the steel used has been added to Table 1 in the paper.

Material

C

Mn

Si

P

S

Ni

O

Ce

Fe

Q345R

0.14

1.45

0.27

0.012

0.004

0.07

0.06

1.59

Bal

H10Mn2

0.90

1.82

0.04

0.011

0.005

<0.10

<0.10

2.72

Bal

H10Mn2+SJ101

0.069

1.63

0.29

0.019

0.0058

0

0.0044

1.699

Bal

In Figure 1, submerged arc welding is carried out in two passes. The heat input has been expressed in "kJ/mm" according to the standard. Other issues are also revised in the original text.

Results and Dicussion:

- I cannot find any scientific discussion here. You have not compared your results with other scientific papers. It allows to underline the biggest advantages of your work, which could be used to mark the novelty and necessity of your work. Please add this kind of comparison.

- "Widmanstatten" - please change to proper name: "Widmanstätten"

- Have you observed any welding imperfections? If not, please underline this issue in the text. It proved that proposed parameters were good from technological point of the view.

- What kind of hardness scale was used, HV5, HV10...? Please mark in the paper.

- Fig. 13 - please show the most important values.

- Table 4- the name of table is "Impact toughness". However, I cannot find impact strength in the table.

R: Thanks to the reviewer's suggestion.The importance and novelty of this research is reflected in the research on the microstructure and mechanical properties of the weld zone, heat affected zone and base metal zone after welding under the influence of different heat inputs, so as to find the best heat input and improve the quality after welding.There are six places in the article where Widmanstaten has been changed into the proper name of Widmanstaten ä tten. This article still needs to be improved, without considering the defects. The standard used for hardness is Vickers hardness.The maximum hardness of welded joint 1#, 2#, 3# and 4# are 226HV, 265HV, 272HV and 231HV, respectively. The maximum hardness of welded joint decreases significantly with the increase of welding heat input and welding current.

Conclusions:

- Please support conclusions with the quantitative results.

R:Thanks to the reviewer's suggestion. Relevant quantitative results have been added to the conclusion.

Round 2

Reviewer 2 Report

The authors made efforts to address the reviewer's comments. It can be published now.

Reviewer 3 Report

Dear Authors,

Thank you for your response. The paper has been improved. I fully recommand it for publishing.